biomechanics/biophysics/evolution

bio-inspired adhesion, Dactyloidae, epidermal outgrowths, Gekkota, microstructures, spinules

**Author for correspondence:**
Austin M. Garner
e-mail: amg149@uakron.edu

# Revisiting the classification of squamate adhesive setae: historical, morphological and functional perspectives

## Austin M. Garner[1] and Anthony P. Russell[2]

[1]Integrated Bioscience Program, Department of Biology, The University of Akron, Akron, OH 44325-3908, USA
[2]Department of Biological Sciences, University of Calgary, Calgary, Alberta, Canada T2N 1N4

(iD) AMG, 0000-0003-1053-9168

Research on gecko-based adhesion has become a truly interdisciplinary endeavour, encompassing many disciplines within the natural and physical sciences. Gecko adhesion occurs by the induction of van der Waals intermolecular (and possibly other) forces between substrata and integumentary filaments (setae) terminating in at least one spatulate tip. Gecko setae have increasingly been idealized as structures with uniform dimensions and a particular branching pattern. Approaches to developing synthetic simulacra have largely adopted such an idealized form as a foundational template. Observations of entire setal fields of geckos and anoles have, however, revealed extensive, predictable variation in setal form. Some filaments of these fields do not fulfil the morphological criteria that characterize setae and, problematically, recent authors have applied the term 'seta' to structurally simpler and likely non-adhesively competent fibrils. Herein we briefly review the history of the definition of squamate setae and propose a standardized classificatory scheme for epidermal outgrowths based on a combination of whole animal performance and morphology. Our review is by no means comprehensive of the literature regarding the form, function, and development of the adhesive setae of squamates and we do not address significant advances that have been made in many areas (e.g. cell biology of setae) that are largely tangential to their classification and identification. We contend that those who aspire to simulate the form and function of squamate setae will benefit from a fuller appreciation of the diversity of these structures, thereby assisting in the identification of features most relevant to their objectives.

# 1. Introduction

Gecko adhesion is a truly interdisciplinary research area, spanning several subdisciplines of biology and extending to many areas of physical science. Adhesively competent squamates, those taxa with the ability to support their body weight on vertical, low-friction substrata, effect attachment using series of expanded subdigital scales, termed scansors or lamellae (figure 1*a*). These scales carry high aspect ratio, fibrillar outgrowths, most commonly referred to as setae (figure 1*b*), terminating in at least one triangular-shaped tip, known as spatulae [1] (figure 1*c*). The mechanisms involved in squamate attachment have been debated for decades and proposed agencies have included glue-like secretions, suction, electrostatic interactions, mechanical interlocking and intermolecular forces (e.g. van der Waals and/or capillary interactions) (reviewed in [2,3]). Adhesive secretions [4], suction [5,6] and mechanical interlocking [7] were ruled out early on, and despite work showing that capillary and/or electrostatic interactions could be important sources of attachment in squamates [8–10], most contemporaneous studies of squamate adhesion cite van der Waals forces as the primary mechanism of attachment, stemming from the results of two seminal studies published in the early 2000s [11,12].

Autumn *et al.* [11,12] recorded the adhesive forces and mechanics of isolated gecko setae and suggested that squamate attachment largely occurs via van der Waals intermolecular forces induced when shear (parallel) forces reorient the spatular faces to make intimate contact with the substratum's surface. These discoveries served as inspiration for the development of the next generation of reversible synthetic adhesives through mimicking the functional properties of the gekkotan adhesive apparatus [13–16]. Autumn *et al.* [11,12], however, used setae from a single species of gecko, the Tokay gecko (*Gekko gecko*) which exhibits a rather complex manifestation of the gekkotan adhesive apparatus [17], and therein reported generalized statements regarding the form and dimensions of gekkotan setae. The interdisciplinary research that followed these studies adopted this view, unwittingly leading to a widespread inference that gekkotan setae exhibit uniform dimensions and a particular branching pattern, despite early and contemporaneous work describing considerable intraindividual and interspecific variation in setal dimensions and form [1,18–24]. These limited views of the form of squamate setae not only obscure the actual variation naturally occurring, but also undermine the value that this variation could have for developing simulacra.

Concurrently, at the other extreme of the variational continuum, some authors [25,26] have applied the term 'seta' to structurally simpler outgrowths lacking spatulate tips (e.g. figure 1*d*). The outermost layer of the epidermis (Oberhäutchen) of most squamates bearing adhesive subdigital pads is adorned with short, spine-like outgrowths, generally referred to as spinules [18,19,27–30]. Various studies have shown that the subdigital surface of adhesively competent squamates exhibits a continuum of epidermal outgrowth form including spinules, various intermediate morphotypes (e.g. figure 1*d*), and fully elaborated filaments with spatulate tips [1,18,20,31,32]. Thus far, however, only fibrils bearing at least one spatulate tip have been demonstrated to generate sufficient adhesive force to support whole animal static clinging and locomotion on vertical, low-friction substrata [33,34]. Therefore, recent efforts redefining what constitutes a squamate seta cloud the variability present in both form and function of squamate epidermal fibrillar outgrowths.

Herein we briefly review the etymology of the word 'seta', its usage in the zoological sciences, and the history of research describing squamate adhesive setae. We then detail a number of studies that highlight the immense variability in form and dimensions of squamate setae and also those contributions that have introduced substantial confusion to their definition and classification. We end this contribution by proffering an operationally standardized definition of setae to distinguish them from other epidermal outgrowths and also provide guidance on the classification of non-setal outgrowths. By so doing, we highlight the natural variation in the form and function of squamate epidermal outgrowths, providing researchers who aspire to replicate the functional properties of setae with a broader spectrum from which to select those most pertinent to their specific objectives.

# 2. Etymology of the word 'seta' and its usage in the zoological sciences

The word 'seta' is Latin in origin (*sēta*, *saeta*) and the Oxford English Dictionary defines it as 'a bristle; bristle-like appendage' [35]. Although also used in the botanical sciences, this term is used extensively in zoology, particularly in reference to structures present in various invertebrate groups (e.g. annelid worms, arthropods). In adhesively competent hexapods and arachnids, the term 'seta' is used to refer to the adhesive structures present on their distal appendages, which can be covered in various regions by additional, smaller adhesive outgrowths, known as microtrichia or setules (Latin meaning 'small seta or bristle' [36]) [37–41].

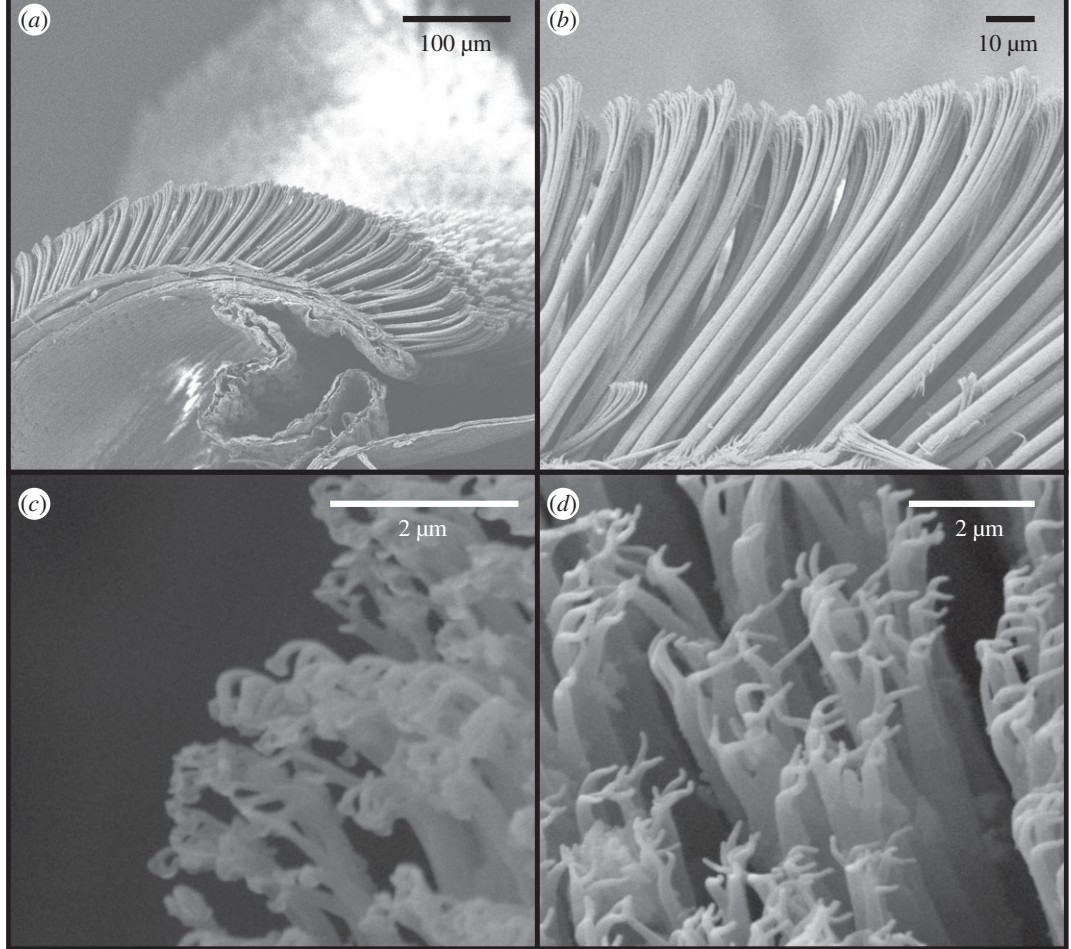

**Figure 1.** Subdigital epidermal fibrillar outgrowths of the Tokay gecko (*Gekko gecko*). (*a*) Expanded scales (scansors). (*b*) Elaborated, branched outgrowths (setae) (*c*) Triangular-shaped tips (spatulae) (*d*) Another form of subdigital epidermal outgrowth—branched prongs.

The setae and microtrichia of hexapods and arachnids vary greatly in form across and within species, and the classification and naming of such structures is often related to this. Three types of setae, for example, have been identified in beetles: pointed, spatulate (possessing expanded, triangular-shaped tips) or discoidal (possessing expanded discs; only observed in male beetles) [37,42]. Similar to its usage in hexapods and arachnids and representing putatively analogous but non-homologous structures, the term 'seta', in reference to vertebrates, is most often attributed to the outgrowths of some geckos and anoles that permit effective whole animal adhesion and locomotion on vertical, low-friction substrata [1]. Such outgrowths invariably possess at least one expanded, triangular-shaped tip, known as spatulae [1,33,34]. As we will discuss below, recent work has attempted to expand the employment of the word 'seta' in squamates and in so doing has added substantial confusion regarding their identification and recognition.

We now move on to present a brief review of how squamate setae were discovered and how various publications have referred to their overall form and dimensions. It is important to note that our review is by no means exhaustive of the literature concerning the form, function and development of squamate setae, nor does it address significant advances that have been made with regard to other attributes (e.g. cell biology of setae [43–46]) that we consider tangential to the classification and identification of squamate setae and other epidermal fibrillar outgrowths.

## 3. A brief history of the discovery and recognition of squamate setae

### 3.1. Pre-1965: naked eye and light microscopic observations

Early observations of the structural agents of gecko attachment were mostly made using the naked eye or light microscope and little in the way of fine structure could be ascertained. In 1816, Home [47,48]

recounted Joseph Banks's observations of geckos scaling smooth, vertical walls and provided an account of the external appearance of gecko subdigital pads, noting that they are split into 'transverse slits', the edge of each being 'serrated' [47] and carrying complex 'rows of a beautiful fringe' [48]. Over a century later, Hora [49] stated that 'lamellae bear innumerable, branching setose processes (or hair-like excrescences)', thus being one of the first to use a variant of 'seta', this being adapted from German descriptions such as 'cuticularborsten' (cuticular bristles) [50], and reflecting the widespread use in zoology of 'seta' to refer to a stiff hair or bristle. Until 1965, various similar terms were used for the structural agents of gecko attachment, including bristles [7,50,51] and brushes [6,51,52]. This nomenclature was consistent with the prevailing view that gecko setae were functionally stiff structures that engaged with the substratum in a similar manner to that of claw tips, although at a much smaller scale [6].

## 3.2. 1965–2000: scanning electron microscopic observations

In 1965, Ruibal & Ernst [1] published the first successful scanning electron microscopic observations of the elongated, spatula-bearing subdigital outgrowths of geckos and anoles and specifically termed them 'setae', thereby establishing criteria, based on their overall dimensions and morphological features, for distinguishing gecko and anole setae from other epidermal filaments. Subsequent studies throughout the 1970s–1990s largely adopted the aforementioned definition of setae and advocated them as the structural agents of gecko attachment [17–19,30,31,53–59]. Although suggested earlier by Haase [60], the scanning electron microscopic observations of Ruibal & Ernst [1] ultimately led Hiller [55,61] to reconsider whether intermolecular forces were the primary mode of adhesion in squamates and examine gecko adhesion on substrates varying in surface energy. Hiller's [55,61] findings that adhesion increased as a function of surface energy lent credence to the idea that intermolecular forces were involved in squamate attachment, although the specific interactions could not be identified at the time.

## 3.3. 2000 till present day: the interdisciplinary phase

Autumn et al. [12] later corroborated Hiller's hypothesis that intermolecular forces governed gecko attachment and provided evidence that van der Waals interactions were the predominant forces involved. Subsequently, other studies have suggested that capillary and electrostatic interactions are also prominent [8–10], but whether such forces operate concurrently with van der Waals interactions and under what circumstances particular forces may dominate has received little attention (but see [10]).

With regard to setal structure and dimensions, Autumn et al. [11] noted that, 'a gecko's foot has nearly five hundred thousand keratinous hairs or setae. Each 30–130 µm long seta is only one-tenth the diameter of a human hair and contains hundreds of projections terminating in 0.2–0.5 µm spatula-shaped structures.' This description was employed and extensively repeated by those seeking to model and exploit the mechanism of gecko attachment but was seldom accompanied by additional direct observations. For example, Gao et al. [62] furnished the following description: 'A gecko is found to have hundreds of thousands of keratinous hairs or setae on its foot; each seta is 30–130 µm long and contains hundreds of protruding submicron structures called spatulae.' Many authors simplified this further, reporting only a single length (generally around 110 µm) and diameter (in the region of 5 µm) for gecko setae [63–68]. Such statements have tended to foster an idealized and potentially invariant structure of gecko setae. Furthermore, although many authors refer to setae as 'keratinous', recent work has revealed that they are actually composed of corneous beta proteins [43,44].

# 4. Transects of squamate setal fields and revelation of the variability of setal dimensions and form

Counter to the view that gecko setae can be characterized by idealized and somewhat invariant form and dimensions, several studies examining transects of entire setal fields of multiple species of geckos (and anoles) have revealed that setal morphology and dimensions vary enormously both between species and within individuals. Employing scanning electron microscopy, Ruibal & Ernst [1] noted that the setal dimensions of geckos and anoles vary proximodistally along the subdigital pad, noting that gecko setae increase in length proximodistally along a scansor and that anoline setae are longest in intermediate portions of lamellae and become shorter more proximally and distally. They also noted ranges of setal length, setal diameter and spatula width for a number of gecko and anole species and provided estimations of setal density. Maderson [19] added to the data collected by Ruibal & Ernst [1]

by including maximal setal length data for a number of skinks, as well as additional species of geckos and anoles, and noted that most species exhibit variation in setal length. Peterson & Williams [31] and Peterson [18] further inspected the subdigital microstructure of several species of *Anolis* and described a transitional series of epidermal outgrowths from spinules to setae through intermediate morphotypes on individual lamellae. Russell *et al*. [20] surveyed setal field transects of the Tokay gecko (*Gekko gecko*) and demonstrated that it also exhibits a similar transitional series of epidermal outgrowths, with different scansors exhibiting different combinations of forms. Additional studies surveying setal field configuration of other gecko and anole species affirm the presence of considerable, yet predictable, dimensional and structural variation of setae and other filaments along the proximodistal axis of the digit [20,21,23,24,32,69].

The extent of variability of setal morphometrics present in individual species of gecko is demonstrated by comparing setal transect data for *Gekko gecko* [20,32] and *Chondrodactylus* (formerly *Pachydactylus*) *bibronii* [22] with the ostensibly species-specific dimensions presented for eight gecko species [70] (figure 2*a* and table 1). These data clearly show that there is no typical seta. Almost all metrics of setal morphology of the two species represented by the transect data overlap with the combined range spanned by the eight exemplar species [70]. Revelation of the actual variability of setal morphology within species has enhanced our understanding of how entire batteries of setae, and the associated and integrated adhesive system, operate during stationary attachment and adhesive locomotion on real-world substrates [20,21,23,24,32,69]. Researchers inspired by squamate attachment systems should take such variability into consideration when designing and fabricating synthetic adhesives to determine the simplest configuration suited to their objectives.

# 5. Inconsistencies and confusion regarding the classification of squamate setae and other epidermal outgrowths

Recently, substantial confusion has been introduced to the understanding of what actually defines a squamate seta. Examination of subdigital epidermal outgrowth morphology of several species of narrow-toed geckos [25] resulted in aspect ratio (fibril length/diameter greater than 10), rather than tip structure and adhesive functionality, being employed as the criterion for distinguishing between setae and other epidermal outgrowths. The stated justification for this was that structures with such aspect ratios are probably capable of enhancing frictional interactions via increased conformation with the substratum [25]. Frictional enhancement, however, is a rather vague descriptor of function when its magnitude is not specified. Epidermal outgrowths on the subdigital surface of chameleon digits have been demonstrated to enhance frictional interactions with the substratum, but not sufficiently so to enable whole animal adhesion and locomotion on low-friction substrata [71]. Thus, this expanded definition of setae [25] decreases the utility of the term by divorcing it from adhesive competence [1] and replacing it with an unspecified level of enhancement of frictional interaction.

Seemingly building upon the aforementioned extended definition of setae [25], Koppetsch *et al*. [26], based upon their examination of the epidermal outgrowths of three species of eublepharid gecko, indicated that all epidermal filaments should be recognized as setae, regardless of aspect ratio. By so doing, they implied, by default, that all of the spinules covering the entire body surface of geckos [19] are setae. The criterion of whether or not such setae are branched was used as the basis for recognition of 'one-unit setae' for unbranched structures and 'multi-unit setae' for branched ones [26]. Such a classificatory scheme introduces further confusion because it employs neither form nor function as identifying criteria for setae and renders comparison between the adhesive filaments of geckos and anoles more difficult (see below).

# 6. A standardized classification scheme of squamate setae and other epidermal outgrowths based on function and form

Inconsistencies regarding the identification of squamate setae and other epidermal outgrowths [1,11,13,25,26] necessitate the development of a classification scheme that is open to modification as more is learned about these structures and their properties. We posit that such a classification scheme should: (i) appropriately reflect the variability in form and function of squamate epidermal outgrowths and (ii) minimize redundancies or ambiguities by building upon empirically based demonstrations of function.

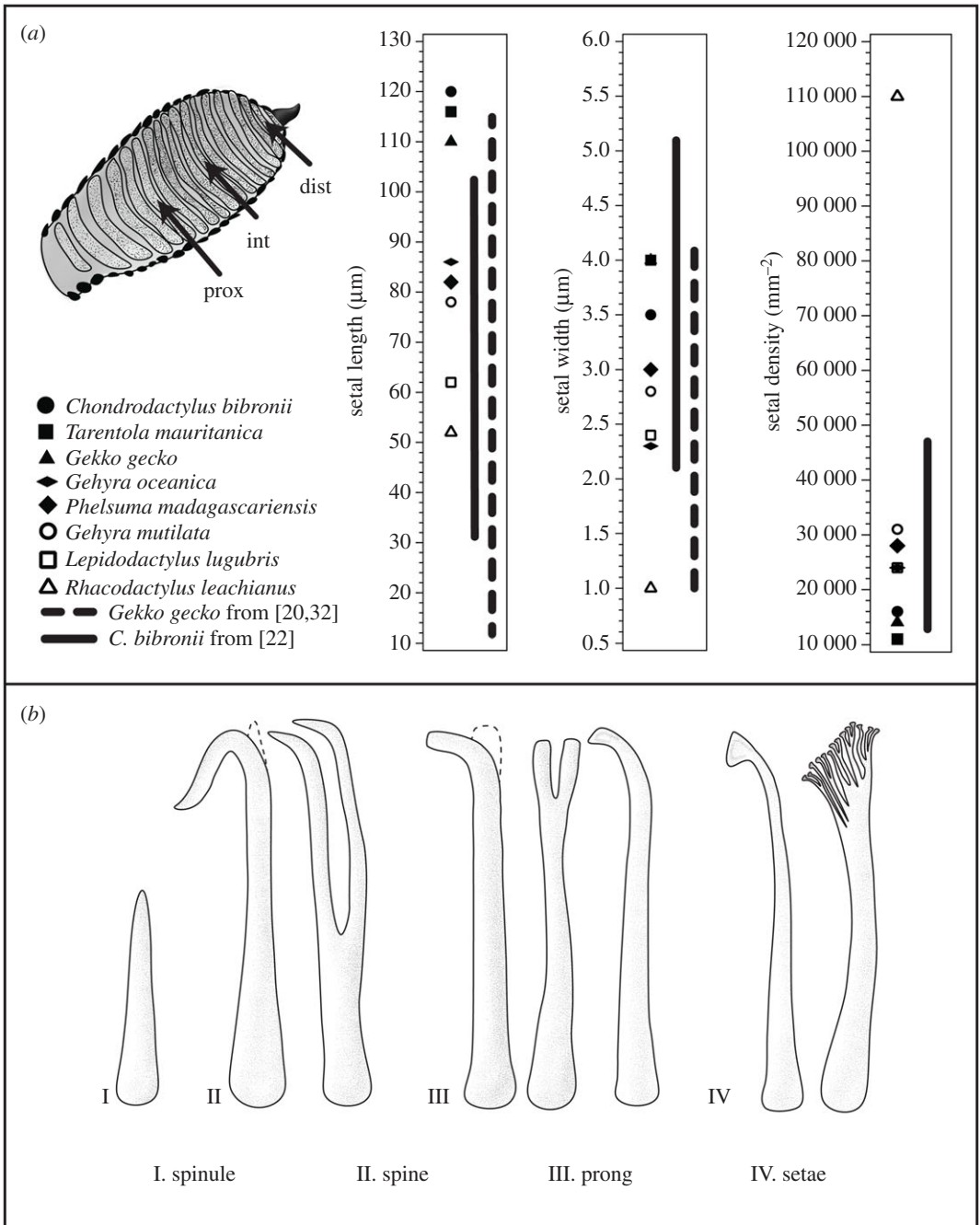

**Figure 2.** (*a*) 'Essential setal morphology' values reported by Peattie [70] plotted against data collected from studies examining setal morphometrics along the proximodistal axis of the digits of both *Chondrodactylus bibronii* [22] and *Gekko gecko* [20,32]. The schematic of a gecko digit at the left depicts the pad regions selected for measurements of setal morphometrics. prox, proximal; int, intermediate; dist, distal. (*b*) Schematic of the variability in the overall form of squamate epidermal outgrowths and corresponding definitions by our classification scheme. Not drawn to scale.

We begin by specifically defining what we believe constitutes a squamate seta. Many taxa have been demonstrated to be adhesively competent as whole organisms, with the ability to cling to and move on vertical, low-friction substrata (e.g. [34,72]). Such taxa have invariably been shown to possess fields of elongate filaments (either branched or unbranched) bearing at least one spatulate tip [1,18,30,33,53,54]. To our knowledge, there are no accounts of taxa with fields of epidermal outgrowths devoid of spatulate tips that can generate sufficient adhesive force to enable the aforementioned locomotor feats. Russell *et al.* [33] documented a transitional epidermal outgrowth series in the sphaerodactylid genus *Gonatodes* and predicted, on the basis of epidermal filament morphology, that *G. humeralis*, but no

**Table 1.** Minimum and maximum values of setal length, width and density of *Chondrodactylus bibronii* [22] and *Gekko gecko* [20,32].

| species | pad region | setal length (µm) | | setal width (µm) | | setal density (mm$^{-2}$) | |
|---|---|---|---|---|---|---|---|
| | | minimum | maximum | minimum | maximum | minimum | maximum |
| *C. bibronii* | proximal | 31 | 68 | 2.1 | 4.3 | 23 333 | 47 333 |
| | intermediate | 36 | 79 | 2.5 | 5.1 | 13 389 | 19 177 |
| | distal | 51 | 103 | 2.4 | 4.8 | 13 278 | 16 117 |
| *G. gecko* | proximal | 12 | 35 | 1 | 1.6 | — | — |
| | intermediate | 41 | 87 | 1.9 | 3.4 | — | — |
| | distal | 80 | 115 | 3.6 | 4 | — | — |

**Table 2.** Our proposed classificatory system of squamate epidermal fibrillar outgrowths, including terms, definitions, length criteria and known morphological variants.

| outgrowth class | description | length (µm) | potential morphological variants |
|---|---|---|---|
| spinules | short, tapered outgrowths that cover the majority of the epidermis of squamates possessing a spinulate oberhäutchen (outermost layer of the epidermis) | <1 | — |
| spines | elongated outgrowths with pointed/tapered ends | >1 | hooked; branched |
| prongs | elongated outgrowths with blunt ends | >1 | hooked; branched; flattened, but not spatulate tips (previously termed the 'seta-prong' intermediate in anoles [17]) |
| setae | outgrowths possessing at least one spatulate tip that are collectively capable of adhesively supporting the whole animal on vertical, low-friction substrata during static clinging and locomotion (as per the empirical tests outlined above) | >1 | branched |

congener, would prove to be adhesively competent because it alone bears filaments with spatulate tips. This was later demonstrated by Higham *et al.* [34], suggesting that whole animal adhesion is only possible when spatulate-tipped epidermal outgrowths are present. Therefore, such structures may reasonably be regarded as setae. We recognize that similar investigations have not been repeated for other taxa, and thus it remains possible that certain outgrowths lacking spatulate tips may be capable of generating considerable adhesive interactions, but such needs to be demonstrated empirically before they can be considered as setae and the concept of setae expanded accordingly. Empirical validation of adhesive competence of epidermal filaments could be demonstrated by: (i) directly observing whole animal static clinging and locomotion on vertical, low-friction substrata and/or (ii) measuring adhesive force capacity of individual fibres that, when scaled appropriately (density per unit area versus body mass), could cumulatively support whole animal static clinging and locomotion on vertical, low-friction substrata.

Squamate epidermal outgrowths not capable of supporting effective attachment and locomotion should be referred to on the basis of the form of their tips, as has historically been the approach for both squamates [1,18,27–31] and invertebrates [37–42]. We present terms, definitions and sketches of four classes of epidermal outgrowths to accommodate the known variability in their form in table 2 and figure 2*b*. We also provide general length criteria which allow spinules, the short, tapered outgrowths that adorn the epidermal surface of some squamates, to be distinguished from the various

classes of elongate epidermal outgrowths. We posit that form should be the primary arbiter of these classes (with the exception of setae which also includes a functional arbiter) because of the overlap in dimensions that exists (or potentially exists) between outgrowths. All observed epidermal filaments should be able to be placed into one of our descriptive categories. Novel nomenclature should be introduced only when the classificatory system cannot accommodate filament form and/or function.

# 7. Conclusion

A deluge of studies investigating squamate adhesion flowed from the demonstration by Autumn and colleagues [11,12] that elongate, spatulate outgrowths on the subdigital surface of geckos are responsible for adhesion via van der Waals interactions. These seminal studies prompted investigations far beyond the confines of conventional biology and seemingly unwittingly led to a limited, stereotypical view of adhesively competent setae, thereby potentially constraining the scope of studies seeking to understand and replicate this 100-million-year-old [73] naturally occurring nanotechnology. All gekkotan setae so far examined, by virtue of being multiply branched, are more complex than those of anoles. Gecko setae, however, may be considerably simpler than the frequently stated exemplar that is '30–130 μm long and contains hundreds of protruding submicron … spatulae' [62]. Concomitantly, alternative definitions of what constitutes a seta were advanced that have obfuscated the classification of squamate epidermal outgrowths. In an attempt to remedy these situations, we introduce a standardized classification scheme that restricts outgrowths classified as setae to those structures empirically demonstrated to support whole animal static adhesion and locomotion on vertical, low-friction substrata. Currently, only elongated fibrils with at least one spatulate tip satisfy this criterion, thereby constituting our working definition of a squamate seta. The remaining epidermal outgrowths of squamates should be classified on the basis of their overall form. The concept of 'seta' can be expanded to accommodate one or more of them if they can be empirically demonstrated to support whole animal static adhesion and locomotion on vertical, low-friction substrata. The naturally occurring variability in the form and function of squamate fibrillar outgrowths can be an important source of inspiration for those seeking to replicate their functional properties.

Data accessibility. This article has no additional data.

Authors' contributions. A.M.G. and A.P.R. developed the ideas contained within the manuscript, drafted initial versions of the text, revised the manuscript, and approved the final version for publication. A.M.G. generated the figures.

Competing interests. We declare we have no competing interests.

Funding. A.P.R. acknowledges financial support from a Natural Science and Engineering Research Council of Canada Discovery (grant no. 9745-2008).

Acknowledgements. We thank Travis Hagey and two anonymous reviewers for comments that greatly improved the manuscript.

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
