## [Peer Review File · Royal Society Open Science]

Review History

RSOS-202039.R0 (Original submission)

Review form: Reviewer 1

Is the manuscript scientifically sound in its present form?

Yes

Are the interpretations and conclusions justified by the results?

Yes

Is the language acceptable?

Yes

Do you have any ethical concerns with this paper?

No

Have you any concerns about statistical analyses in this paper?

No

Recommendation?

Accept with minor revision (please list in comments)

Comments to the Author(s)

See attached file (Appendix A).

Decision letter (RSOS-202039.R0)

Dear Mr Garner

On behalf of the Editors, we are pleased to inform you that your Manuscript RSOS-202039 "Revisiting the classification of squamate adhesive setae: historical, morphological, and functional perspectives" has been accepted for publication in Royal Society Open Science subject to minor revision in accordance with the referees' reports. Please find the referees' comments along with any feedback from the Editors below my signature.

Please submit your revised manuscript and required files (see below) no later than 7 days from today's (ie 11-Dec-2020) date. Note: the ScholarOne system will 'lock' if submission of the revision is attempted 7 or more days after the deadline. If you do not think you will be able to meet this deadline please contact the editorial office immediately.

on behalf of Professor Emily Standen (Associate Editor) and Kevin Padian (Subject Editor)
openscience@royalsociety.org

Associate Editor Comments to Author (Professor Emily Standen):

Dear Dr. Garner,

We have received a short but very clear review that makes a few helpful suggestions for clarifying your text. These are very minor changes but I think will make a nice improvement to the presentation of the ms. I hope that you agree.

I look forward to seeing this paper again for as fast a turn around as we can manage.

EMS

Reviewer comments to Author:

Reviewer: 1

Comments to the Author(s)

see attached file

===PREPARING YOUR MANUSCRIPT===

Your revised paper should include the changes requested by the referees and Editors of your manuscript. You should provide two versions of this manuscript and both versions must be provided in an editable format:
one version identifying all the changes that have been made (for instance, in coloured highlight, in bold text, or tracked changes);
a 'clean' version of the new manuscript that incorporates the changes made, but does not highlight them. This version will be used for typesetting.

===PREPARING YOUR REVISION IN SCHOLARONE===

Author's Response to Decision Letter for (RSOS-202039.R0)

See Appendix B.

Decision letter (RSOS-202039.R1)

Dear Mr Garner,

It is a pleasure to accept your manuscript entitled "Revisiting the classification of squamate adhesive setae: historical, morphological, and functional perspectives" in its current form for publication in Royal Society Open Science.

You can expect to receive a proof of your article in the near future. Please contact the editorial office (openscience@royalsociety.org) and the production office (openscience_proofs@royalsociety.org) to let us know if you are likely to be away from e-mail contact – if you are going to be away, please nominate a co-author (if available) to manage the proofing process, and ensure they are copied into your email to the journal.

on behalf of Professor Emily Standen (Associate Editor) and Kevin Padian (Subject Editor)
openscience@royalsociety.org

Appendix A

Revisiting the classification of squamate adhesive setae: historical, morphological, and functional perspectives

I feel the additions made by the authors in this updated version of the manuscript greatly strengthen the paper.

I only have two suggestions. Throughout the manuscript, the authors refer to “adhesively competent” squamates. I’m assuming this is referring to species that do or do not generate adhesion, ie negative normal force with their toe pads? Since part of the authors definition of setae allude to performance, I’d suggest they define “adhesively competent” early in the manuscript to differentiate the types of species they’re talking about vs other species with similar structures that also presumably aid in climbing like chameleons. The authors mention this in lines 228-229.

Line 284-296: I think this bulleted list is great but may be better as a table. Do the authors have any reasoning for assigning spines an upper length of 15um and prongs an upper length of 20um? I might suggest simplifying the length suggestions to be simplified to be over or under 1 um. I would also suggest adding a note that spines come to a “pointed or tapered tip” whereas prongs have a blunt tip, and setae have a flattened and widened tip.

In addition to listing suggested lengths, I think the authors could also propose aspect ratio ranges for each class of structure. I suggested aspect ratios of less than 2x tall as wide for spinules and more than 2x tall as wide for everything else, but the authors should feel free to use whatever values they think are appropriate. This defined aspect ratio would provide a concrete definition of “elongated.”

I think something like the below table would be a great improvement

	Description	Length	Aspect Ratio	Variations
Spinule	short, tapered outgrowths that cover the majority of the epidermis of squamates possessing spinulate Oberhäutchen (outer most layer of the epidermis)	< 1 um	Less than twice as tall as it is wide	
Spine	elongated outgrowths with pointed/tapered ends	> 1um	At least twice as tall as it is wide	Can be hooked or branched
Prong	elongated outgrowths possessing blunt ends	> 1um	At least twice as tall as it is wide	Can be hooked, branched, or have flattened (but not widened) ends
Setae	elongated outgrowths with flattened and widened ends (a spatulate tip) that are collectively capable of supporting whole animal static adhesion and locomotion on vertical, low friction substrata	> 1um	At least twice as tall as it is wide	Can be branched

The authors could also mention that the categorization of elongate outgrowths based on the structure’s tip (spines vs prongs vs setae) is complementary to similar setal categories observed in beetles and so these categories have precedence and are consistent and generalizable.

Otherwise, I’m glad I could contribute!

-Travis Hagey

Appendix B

Associate Editor Comments to Author (Professor Emily Standen):

Dear Dr. Garner,

We have received a short but very clear review that makes a few helpful suggestions for clarifying your text. These are very minor changes but I think will make a nice improvement to the presentation of the ms. I hope that you agree.

I look forward to seeing this paper again for as fast a turn around as we can manage.

EMS

Thank you, Professor Standen. We respond to each of Dr. Hagey's comments below.

Reviewer:1

Comments to the Author(s):

I feel the additions made by the authors in this updated version of the manuscript greatly strengthen the paper.

We thank Dr. Hagey for reviewing our manuscript again and providing additional, helpful insight.

I only have two suggestions. Throughout the manuscript, the authors refer to “adhesively competent” squamates. I’m assuming this is referring to species that do or do not generate adhesion, ie negative normal force with their toe pads? Since part of the authors definition of setae allude to performance, I’d suggest they define “adhesively competent” early in the manuscript to differentiate the types of species they’re talking about vs other species with similar structures that also presumably aid in climbing like chameleons. The authors mention this in lines 228-229.

The term “adhesively competent” refers to those organisms that are capable of supporting their body weight on vertical, low friction substrata. We have provided a definition of this at its first use (lines 49-50).

Line 284-296: I think this bulleted list is great but may be better as a table. Do the authors have any reasoning for assigning spines an upper length of 15um and prongs an upper length of 20um? I might suggest simplifying the length suggestions to be simplified to be over or under 1 um. I would also suggest adding a note that spines come to a “pointed or tapered tip” whereas prongs have a blunt tip, and setae have a flattened and widened tip.

We agree and have converted the bulleted list into a table.

As we discussed in the our second revision of the manuscript, the “range of lengths [of the listed outgrowths] provides some indication of the general differences in this parameter between the outgrowth forms and should only be used as a guide. We posit that form should be the primary arbiter of the various outgrowth classes described below (with the exception of setae which also includes a functional arbiter) because of the overlap in dimensions that exists (or potentially exists) between outgrowths.”. Length ranges were included based on the comments provided by former Reviewers 1 and 2 of the original submission. Dr. Hagey’s suggestion to simplify the length ranges is an excellent idea and we have made that adjustment in the revised manuscript (Table 2).

In addition to listing suggested lengths, I think the authors could also propose aspect ratio ranges for each class of structure. I suggested aspect ratios of less than 2x tall as wide for spinules and more than 2x tall as wide for everything else, but the authors should feel free to use whatever values they think are appropriate. This defined aspect ratio would provide a concrete definition of “elongated.”

Similar to providing ranges of lengths, we are hesitant to declare aspect ratios for each of the classes of fibrillar outgrowths because of the existing (and potentially undiscovered) variability in dimensions. A length cut-off is certainly helpful for differentiating between spinules and spines, but the proposed aspect ratio categories could be problematic given our current classification scheme. For example, an outgrowth that is less than 1 μm in length and has an aspect ratio (length/width) greater than 2 would not fit within any of the categories proposed. We contend that length can be used to differentiate between spinules and spines, but that the rest of the outgrowths should be categorized based upon form alone (with the exception of setae which include a functional arbor).

I think something like the below table would be a great improvement

	Description	Length	Aspect Ratio	Variations
Spinule	short, tapered outgrowths that cover the majority of the epidermis of squamates possessing spinulate Oberhäutchen (outer most layer of the epidermis)	< 1 μm	Less than twice as tall as it is wide	
Spine	elongated outgrowths with pointed/tapered ends	> 1 μm	At least twice as tall as it is wide	Can be hooked or branched
Prong	elongated outgrowths possessing blunt ends	> 1 μm	At least twice as tall as it is wide	Can be hooked, branched, or have flattened (but not widened) ends
Setae	elongated outgrowths with flattened and widened ends (a spatulate tip) that are collectively capable of supporting whole animal static adhesion and locomotion on vertical, low friction substrata	> 1 μm	At least twice as tall as it is wide	Can be branched

The authors could also mention that the categorization of elongate outgrowths based on the structure’s tip (spines vs prongs vs setae) is complementary to similar setal categories observed in beetles and so these categories have precedence and are consistent and generalizable.

We have mentioned this in the revised manuscript (lines 279-281).

Otherwise, I’m glad I could contribute!

-Travis Hagey